# Timing of parental depression on risk of child depression and poor educational outcomes: A population based routine data cohort study from Born in Wales, UK

**Sinead Brophy**[1]*, **Charlotte Todd**[1], **Muhammad A. Rahman**[2], **Natasha Kennedy**[1], **Frances Rice**[3]

**1** Medical School, Swansea University, Swansea, United Kingdom, **2** Cardiff School of Technologies, Cardiff Metropolitan University, Cardiff, United Kingdom, **3** Division of Psychological Medicine and Clinical Neurosciences, Cardiff University, Cardiff, United Kingdom

* s.brophy@swansea.ac.uk

## Abstract

### Background

Maternal depression is a risk factor for depression in children, though the influence of paternal depression has been less well examined. We examined the association between maternal and paternal depression, and the timing of their depression (before or after the child's birth) and outcomes for the child including incidence of child depression and poor educational attainment.

### Methods

A linked routine data cohort study linking General Practitioner(GP), hospital and education records of young people (aged 0 to 30 years) in Wales. Parental and child diagnosis of depression was identified from GP data. Regression analysis examined the association of maternal and paternal depression with time to diagnosis of depression in the child and odds of attaining educational milestones.

### Outcomes

In adjusted models, the relative risk of offspring developing depression was 1.22 if the mother had depression before the child was born, 1.55 if the mother had depression after the child was born and 1.73 if she had depression both before and after the child was born (chronic depression), compared to those were there was no maternal depression history. For achieving milestones at end of primary school, odds were 0.92, 0.88 and 0.79 respectively. Association of depression in the child was similar if the male living in the household had depression with risk ratios of 1.24 (before), 1.43 (after) and 1.27 (before and after) for child diagnosed depression and 0.85, 0.79 and 0.74 for achieving age 11 milestones.

**Data Availability Statement:** The dataset supporting conclusions from this article is available via the Secure Anonymised Information Linkage (SAIL) databank, which is part of the national e-

health records infrastructure for Wales. 19 For further information on the SAIL databank and enquiries in how to access the data, please visit the SAIL website (http://www.saildatabank.com). The findings can be replicated in their entirety by directly obtaining the data from SAIL and following the protocol in the methods section. The authors did not have any special access privileges that others would not have.

**Funding:** The authors received no specific funding for this work. The infrastructure to enable the study was funded by Health Care Research Wales (https://healthandcareresearchwales.org/) which funded; the National Centre for Population Health and Wellbeing Research (https://ncphwr.org.uk/) enabling the involvement of SB, CT, MAR, TK, the National Centre for Mental Health Wales (https://www.ncmh.info/), which supported the involvement of FR, and the Secure Anonymised Information Linkage Database (https://saildatabank.com/).

**Competing interests:** No authors have competing interests.

## Interpretation

Children who live with a parent who has depression are more likely to develop depression and not achieve educational milestones, compared to children who live with a parent who has a history of depression (but no active depression in child's lifetime) and compared to those with no depression. This finding suggests that working closely with families where depression (particularly chronic depression) is present in either parent and treating parental depression to remission is likely to have long-term benefits for children's mental health and educational attainment.

## Introduction

Depression in a parent is a common and potent risk factor for depression in the child [1, 2] and is also associated with a range of adverse child health and educational outcomes including poorer academic attainment [1, 3–8]. To date, the vast majority of research has focused on the effect of maternal depression on offspring outcomes. Depression in a mother increases the likelihood of offspring depression by 3 to 4 fold on average, with 6 to 10 fold increases in risk reported when maternal depression is severe or recurrent [9]. Depressive disorders are a prominent cause of disability worldwide [10, 11] and often onset in adolescence or early adult life. Early-onset depression (by the 20s) is concerning because numerous studies indicate it is associated with particularly poor outcomes including poor physical and mental health [1, 4, 5, 8], suicide [6, 7] and academic failure [12]. Educational performance at school is itself also associated with future health and economic outcomes [13].

Recent studies utilising data linkage approaches have helped enhance understanding over both the prevalence of maternal mental illness among children and adolescents and the impact of parental depression on education outcomes using large population cohorts in Sweden and the UK. The Swedish based population study found that diagnosis of parental depression (both maternal and paternal) throughout a child's life was associated with worse school performance at age 16 [3]. More recently, Abel et al used a UK based cohort and identified high proportions of children are exposed to maternal illness and called for information on paternal mental health as a public health imperative [14].

Indeed, whilst these linkage studies have given greater insight into the field, questions remain regarding the impact of parental depression on child outcomes. This includes the extent to which paternal depression has an association with child outcomes that is independent of maternal depression, with research reporting that paternal effects on child outcomes may be mediated through maternal depression [15]. Innovative approaches to cohort designs are also informative for teasing apart the effects of timing of parental depression. These natural experimental approaches involve examining the impact of timing of exposure to parental depression (e.g. before or after the child's birth) with the assumption that an environmental exposure effect on offspring is only plausible for exposure during the child's life time [8]. This type of design has been applied to offspring antisocial behaviour but has yet been used to examine the association with other offspring outcomes. Moreover, we are not aware of a study using such an approach to examine offspring depression that includes the peak period of risk for major depressive disorder which occurs between late adolescent and early adult life [15].

It is important to examine the effects of timing of maternal and paternal depression on offspring outcomes because this has implications for prevention and early intervention. Previous research suggest that direct exposure effects to parental depression may have adverse effects on

children's developmental outcomes via mechanisms such as impaired parent-child relationships, exposure to stress as well as children learning pessimistic styles of viewing the world [16]. In contrast, depression is heritable in adults and in young people and passive gene-environment correlation whereby inter-generational transmission via seemingly environmental exposures are driven by inherited mechanisms may also exist [17, 18]. Thus, it is important to assess the extent to which "exposure" to parental depression has a direct environmental impact on children because of the implication for prevention of adverse outcomes in the children of depressed parents. The implication of exposure effects is that proactive treatment of parental depression to remission is likely to have added benefits for the parents' children. Thus, such evidence would be informative for recommendations that identify the offspring of parents with depression as meriting special consideration for preventive and early interventions and informing the development of guidelines as to how this might be achieved [19]. We used data from a national sample of over one million young people aged from 0 to 30 years (year of birth 1987 to 2018) (mean offspring age = 14.92 (standard deviation: 8.75, median 14.40) that includes nearly complete coverage (>90%) of primary care records for the population of Wales, UK. Primary care is the setting in the UK where the vast majority of depression is diagnosed and managed [20]. We examined diagnosis of depression and poor academic attainment as outcomes in young people. Diagnosis of parental depression (maternal and paternal) were the exposure variables. We stratified by timing of exposure to parental depression (e.g. before the child was born and in the lifetime of the child) compared to families which had no previous history of depression. The assumption being that only parental episodes during the child's life time would plausibly be expected to impact adversely on the rearing environment [21]. We investigated the impact of parental depressive episodes on their children's depression and educational outcomes in order to investigate the effect of episodes prior to and during the child's lifetime. This study is unique in that it uses this approach to examine outcomes in offspring during their peak period of risk for depression including both maternal and paternal influence.

## Objectives

We sought to assess:

1. Evidence for exposure effects of maternal and paternal depression on offspring depression and educational outcomes.

2. The magnitude of association with offspring outcome similar depending on whether the mother or father was depressed.

## Methods

### Study design

**Procedures.** Data linkage for this routine cohort study was carried out through the SAIL (Secure Anonymised Information Linkage) databank in order to inform the design of the Born In Wales Study (Born In Wales—NCPHWR). The SAIL databank contains anonymised health and administrative datasets on the Welsh population with total population coverage for hospital admission and 90% population coverage for general practice data. Datasets utilised in this study included the Welsh Demographic Service Dataset (e.g. registration with a general practice), the Education Attainment dataset, and hospital admission, (Patient Episode Database for Wales (PEDW)). For each of these datasets, each individual is assigned a unique anonymous identifier, known as an Anonymous Linking Field (ALF). This remains the same across

data sources [22]. In order to determine occupiers of the same household and in the case of our study, identify the stable male linked to the child, a similar procedure is followed for anonymising residential information, whereby each dwelling is assigned a Residential Anonymous Linking Field (RALF) [23]. Thus, we identified "fathers" based on a stable male who was currently living with the mother/child (based on their RALF), was of adult age (over 18 years and aged within 10 years of the mothers' age) and who lived in the same household as the mother for a minimum of 1 year before the birth of the child. This is similar to methods employed previously [24] and attempts to overcome the challenge that linkage to paternity data has yet to be fully established [14]. This variable can examine influence of an adult male in the household but will not capture influence of fathers who do not live with the child.

The SAIL databank was queried using SQL (IBM DB2 9.7). S1 Fig details the flow diagram showing how the sample size was arrived at and processes involved. Exposures were: 1. Maternal diagnosis of depression, 2. Paternal diagnosis of depression. Timing of depression was recorded as: no history of depression (0); depression before birth of child only (1); depression after birth of child only (2); depression before and after birth of the child (3). The diagnosis of depression was based on read codes for major depressive episode (either single or recurrent [25]; see S1 Table) all available GP records were used to identify depression before the birth of the child but all participants need to have a minimum of 2 years of GP records before the birth of the child. In the United Kingdom, GP data are coded using read codes, which contain codes for symptoms, diagnosis, treatment and management; data within hospital admission are recorded using ICD-10 codes [26]. The RECORD statement for reporting routinely linked data was followed throughout [27].

Primary outcomes included offspring diagnosis of depression (utilising the read codes in S1 Table) in either the GP or the hospital records, and offspring educational attainment. Educational attainment was dichotomised to capture achieving or not achieving expected national curriculum levels in core subjects at Key Stages 1, 2 and 3 (KS1, KS2 and KS3) which are captured in the educational dataset. Key Stage 1 is completed at age 6/7, Key Stage 2 at age 10/11, and Key Stage 3 at age 13/14; and Key Stages are assigned by summative teacher assessments. An individual was classed as achieving their Key Stage level if they had passed the core subjects (English/Welsh) and mathematics to the accepted national curriculum level. If they did not achieve the accepted level in either of these core subjects, they were assigned as "not achieved".

Confounders aimed to capture factors associated with key study exposure and outcome variables available in the linked dataset. These included: number of house moves in the first five years of the child's life, Townsend quintile (as an index of socio-economic deprivation), child gender, age of mother at birth of the child, child birth weight and gestational age.

## Statistical analysis

Following extraction from the SAIL databank, the dataset was imported into STATA and cleaned to firstly remove any duplicate entries. The authors had full access to the data to create the study population. All variables to be included in the analysis were then cleaned. Gestational age was cleaned so that any gestational age less than 22 or more than 45 weeks were replaced with missing values. Age of mother at birth was cleaned so that any age less than 10 and more than 65 was replaced as a missing value. Birth weight was cleaned so that any weight less than 0.435kg and more than 7kg was replaced with a missing value. Age of depression diagnoses under 5 were also replaced with missing values. Missing values in the dataset on continuous variables such as gestation age or mother age were excluded from analysis. For categorical variables such as key stage achievement, missing variables were given their own category. Numbers of each variable can be seen in Table 1.

Statistical analyses were conducted using Stata version 13. Cox regression analysis was used to examine the relationship between parental depression and outcomes adjusted for confounders (i.e. gender, deprivation, previous attainment). The follow-up was calculated as time to depression diagnosis in the child or censored at date of death or date of end of study (July 2018)/last GP download. Logistic regression was used to explore the influence of maternal and paternal depression on educational outcomes (Key Stage 1, 2 and 3 achieved/not achieved).

### Ethical approval

This study was approved by the Information Governance Review Panel (IGRP) at Swansea University.

## Results

Data from a national sample of young people aged from 0 to 30 years (year of birth 1987 to 2018) and their parents (1,080,118 mother child entries and 369,426 stable male child entries) were included (mean offspring age = 14·92 (standard deviation: 8·75, median 14.40) who resided in Wales, United Kingdom.

### Depression

Tables 1 and 2 show that 34·5% (326,408/945,713) of mother child entries and 18.0% (58,103/323,572) of stable male child entries had a diagnosis of depression respectively. For mothers, 7·7% had a diagnosis before the child was born only (and no depression during the child's lifetime), 20·9% had a diagnosis during the child's lifetime only and 5·9% had a diagnosis both before and during the child's lifetime. For fathers/stable male, the figures were: 4·8%, 11.1% and 2.1% respectively. The mean age at first diagnosis of depression in the mother ranged from 23·5 years (before child born) to 34.6 years (after child born) and for the father, this was 25.1 to 37.9. In the offspring, 4.34% (38,643/890,536) of children had a diagnosis of depression (2.85%: 12,942/454,147 boys and 5.89%:25,700/436,319 girls).

Tables 3 and 4 show associations between maternal depression and offspring depression and attainment without adjustment for confounders. Three groups (before only, after only, before and after) are compared to the reference never depressed group: those where there was no diagnosis of maternal depression on record. The mean age at child depression diagnosis in these groups was 19 years, highlighting the importance of the transition into early adult life for depression risk. For maternal depression, the association with offspring depression diagnosis increased by maternal depression group with the lowest association observed for the group of children whose mothers experienced depression only prior to their birth (HR = 1·32, 95% CI = 1·21, 1·43). For those where maternal depression occurred during the child's lifetime only, the association with offspring depression was significantly higher (HR = 2·00, 95% CI = 1·96, 2·05) and the strongest association was seen in the group of offspring whose mothers experienced depression both prior to and during their lifetime (HR = 2·25, 95% CI = 2·15, 2·35). Importantly the confidence intervals did not overlap for those who had depression prior to child's birth only and the other maternal depression groups where the child was exposed to maternal depression. This is consistent with potential environmental effects of maternal depression on child depression diagnosis. For paternal depression (Table 2), associations with child depression were observed. However, the risk effect on child depression was similar for each of the three paternal depression groups (prior, after and both prior and after child's birth) and the confidence intervals overlapped. A comparison of those who had depression after (but not before the child's birth) and those who had depression only before showed a HR 1.16 (95% CI: 0.95–1.4), showing no significant difference.

**Table 1. Descriptive characteristics of maternal depression, offspring depression and offspring educational attainment.**

| | | Never had depression | Previous history of depression only | Depression after birth | Previous history of depression & diagnosis after birth |
|---|---|---|---|---|---|
| **Overall % of mother child entries (n/N)** | | 65.49 (619,305/ 945,713) | 7.70 (72,828 /945,713) | 20.92 (197,850 /945,713) | 5.89 (55,730 /945,713) |
| **Mean age of first diagnosis of depression in mother (SD)** | | . . . | 23.69 (5.46) | 34·57 (9.01) | 23.54 (5.57) |
| **Mean age became mother (SD)** | | 28.83 (5.79) | 30.12 (5.65) | 26.43 (5.75) | 28.49 (5.75) |
| **% of mothers where stable male identified** | | 35.94 (222,594/ 619,305) | 36.05 (26,255/72,828) | 29.62 (58,596/ 197,850) | 30.39 (16,935/55,730) |
| **% of children with depression (n/N)** | Overall | 2.93 (16,443/ 560,598) | 0.98 (637/65,085) | 9.21 (17,955/194,904) | 4.11 (2,254/54,781) |
| | 95% CI of difference | | -1.95 (-1.86,-2.04) | 6.28 (6.14,6.28) | 1.18 (1.01,1.36) |
| | Girls | 4.02 (11,026/ 274,446) | 1.34 (428/31,936) | 12.44 (11,809/94,961) | 5.54 (1,474/26,614) |
| | 95% CI of difference | | -2.68 (-2.53, -2.82) | 8.42 (8.20,8.64) | 1.52 (1.24,1.81) |
| | Boys | 1.89 (5,417/286,098) | 0.63 (209/33,146) | 6.15 (6,146/99,935) | 2.77 (780/28,165) |
| | 95% CI of difference | | -1.26 (-1.16, -1.36) | 4.26 (4.10,4.42) | 0.88 (0.68,1.08) |
| **Mean age of diagnosis of depression in child (SD)** | | 20.18 (2.80) | 19.04 (3.00) | 19.84 (2.89) | 19.04 (2.99) |
| **Crude Hazard ratio of child depression (95% CI)** | | | 1.32 (1.21–1.43) | 2.00 (1.96–2.05) | 2.25 (2.15–2.35) |
| **%of children achieving at age 6/7 (KS1) (n/N)** | Overall | 86.68 (116,738/ 134,682) | 82.14 (13,898/16,920) | 80.32 (36,424/45,350) | 77.85 (15,254/19,594) |
| | 95% CI of difference | | -4.54 (-3.95, -5.15) | -6.36 (-5.95, -6.77) | -8.83 (-8.22, -9.44) |
| | Girls | 90.61 (59,540/ 65,707) | 86.53 (7,190/8,309) | 85.37 (18,712/21,918) | 83.44 (7,900/9,468) |
| | 95% CI of difference | | -4.08 (-3.3, -4.86) | -5.24 (-4.73, -5.76) | -7.18 (-6.4, -7.97) |
| | Boys | 82.92 (57,191/ 68,968) | 77.90 (6,707/8,610) | 75.59 (17,711/23,431) | 72.62 (7,353/10,125) |
| | 95% CI of difference | | -5.03(-4.12, -5.96) | -7.34 (-6.72, -7.96) | -10.30 (-9.40, -11.22) |
| **% of children achieving at age 10/11 (KS2)(n/N)** | Overall | 86.18 (102,648/ 119,102) | 83.05 (7,569/9,114) | 80.38 (42,120/52,399) | 79.01 (12,036/15,253) |
| | 95% CI of difference | | -3.14 (-2.36, -3.94) | -5.80 (-5.41, -6.20) | -7.28 (-6.61, -7.96) |
| | Girls | 89.20 (51,864/ 58,141) | 87.12 (3,807/4,370) | 84.40 (21,483/25,453) | 83.17 (6,232/7,493) |
| | 95% CI of difference | | -2.09 (-1.09, -3.14) | -4.80 (-4.29, -5.20) | -6.03 (-5.16, -6.93) |
| | Boys | 83.30 (50,776/ 60,952) | 79.30 (3,762/4,744) | 76.59 (20,636/26,945) | 74.99 (5,804/7,740) |
| | 95% CI of difference | | -4.00 (-2.84, -5.22) | -6.72 (-6.14, -7.31) | -8.32 (-7.32, -9.34) |
| **% of children achieving at age 13/14 (n/N)** | Overall | 78.53 (89,544/ 114,021) | 73.84 (4,010/5,431) | 68.19 (39,252/57,565) | 67.66 (7,374/10,898) |
| | 95% CI of difference | | -4.70 (-3.52, -5,91) | -10.35 (-9.90,-10.80) | -10.87 (-9.97,-11.78) |
| | Girls | 82.74 (46,174/ 55,807) | 79.83 (2,078/2,603) | 73.20 (20,617/28,167) | 73.01 (3,855/5,280) |
| | 95% CI of difference | | -2.91 (-1.38. -4.52) | -9.54 (-8.94, -10.15) | -9.73 (-8.51, -10.98) |
| | Boys | 74.50 (43,364/ 58,206) | 68.32 (1,932/2,828) | 63.39 (18,635/29,398) | 62.64 (3,519/5,618) |
| | 95% CI of difference | | -6.18 (-4.46, -7.96) | -11.11 (-10.46, -11.77) | -11.86 (-10.56, -13.18) |

95% CI of difference = the reference group for confidence intervals is mother never had depression.

**Table 2. Descriptive characteristics of stable male depression, offspring depression and offspring educational attainment.**

| | | Never had depression | Previous history of depression | Depression after birth | Previous history of depression & diagnosis after birth | No stable male identified |
|---|---|---|---|---|---|---|
| **Overall % of stable males (n/N)** | | 82.04 (265,469/ 323,572 | 4.79(15,490 /323,572 | 11.06(35,775 /323,572 | 2.11 (6,838 /323,572 | - |
| **Mean age of first diagnosis of stable male (SD)** | | | 25.91 (6.34) | 37.87 (16.33) | 25.06 (5.79) | - |
| **% of children with depression (n/N)** | Overall | 2.86 (6,821/ 238,104) | 0.81 (112/13,872) | 6.81 (2,295/ 33,693) | 2.29 (147/ 6,425) | 4.87 (28,482/ 584,818) |
| | 95% CI of difference | | -2.06 (-1.88, -2.21) | 3.95 (3.67, 4.23) | -0.58 (-0.18, -0.92) | 2.01 (1.92, 2.09) |
| | Girls | 3.91 (4,520/ 115,628) | 1.21 (82/6,774) | 9.01 (1,470/ 16,319) | 2.91 (90/ 3,093) | 6.61 (19,009/ 287,692) |
| | 95% CI of difference | | -2.70 (-2.39, -2.96) | 5.10 (4.65, 5.56) | -1.00 (-0.34, -1.05) | 2.77 (2.55, 2.84) |
| | Boys | 1.88 (2,301/ 122,461) | 0.42 (30/7,098) | 4.75 (825/17,372) | 1.71 (57/3,332) | 3.19 (9,473/ 297,075) |
| | 95% CI of difference | | -1.46 (-1.26, -1.60) | 2.87 (2.55, 3.20) | -0.17 (-0.56, 0.34) | 1.31 (1.21,1.41) |
| **Mean age of diagnosis of depression in child** | | 20.01 (2.88) | 19.23 (2.59) | 19.84 (2.82) | 18.98 (3.46) | 19.93 (2.87) |
| **Crude Hazard ratio of child depression (95% CI)** | | | 1.44 (1.18–1.74) | 1.66 (1.58–1.74) | 1.47 (1.25–1.73) | 1.27 (1.24–1.30) |
| **% of children achieving at age 6/7 (KS1) (n/N)** | Overall | 87.72 (55,070/ 62,778) | 82.74 (2,647/3,199) | 82.57 (7,827/ 9,479) | 79.60 (1,744/2,191) | 83.03 (127,641/ 153,729) |
| | 95% CI of difference | | -4.98 (-3.68, -6.35) | -5.15 (-4.36,-5.97) | -8.12 (-6.47, -9.88) | -4.69 (-4.37, -5.01) |
| | Girls | 91.33 (27,871/ 30,517) | 87.28 (1,352/1,549) | 87.32 (4,110/ 4,707) | 85.38 (888/1,040) | 87.58 (65,472/ 74,753) |
| | 95% CI of difference | | -4.05 (-2.45, -5.83) | -4.01 (-3.01, -5.01) | -5.94 (-3.90, -8.24) | -3.74 (-3.35, -4.14) |
| | Boys | 84.31 (27,198/ 32,260) | 78.48 (1,295/1,650) | 77.89 (3,717/ 4,772) | 74.37 (856/1,151) | 78.72 (62,160/ 78,965) |
| | 95% CI of difference | | -5.82 (-3.87, -7.91) | -6.42 (-5.19, -7.68) | -9.94 (-7.47, -12.57) | -5.59 (-5.10, -6.08) |
| **% of children achieving at age 10/11 (KS2) (n/N)** | Overall | 86.97 (48,575/ 55,855) | 82.52 (1,473/1,785) | 81.23 (8,218/ 10,117) | 78.95 (1,283/1,625) | 83.07 (118,087/ 142,148) |
| | 95% CI of difference | | -4.45 (-2.73,-6.30) | -5.74 (-4.94, -6.56) | -8.01 (-6.08, -10.08) | -3.89 (-3.55, -4.23) |
| | Girls | 89.85 (24,291/ 27,036) | 84.74 (733/865) | 85.63 (4,212/ 4,919) | 84.58 (669/791) | 86.66 (60,201/ 69,465) |
| | 95% CI of difference | | -5.11 (-2.83, -7.68) | -4.22 (-3.20,-5.29) | -5.27 (-2.89, -7.98) | -3.18 (-2.74, -3.62) |
| | Boys | 84.26 (24,281/ 28,816) | 80.43 (740/920) | 77.07 (4,006/ 5,198) | 73.62 (614/834) | 79.64 (57,879/ 72,674) |
| | 95% CI of difference | | -3.83 (-1.36, 6.55) | -7.19 (-5.99, -8.43) | -10.64 (-7.73, -13.76) | -4.62 (-4.11, -5.13) |
| **% of children achieving at age 13/14 (KS3) (n/N)** | Overall | 79.82 (42,046/ 52,674) | 72.74 (758/1,042) | 70.19 (7,187/ 10,240) | 68.78 (802/1,166) | 73.17 (101,875/ 139,238) |
| | 95% CI of difference | | -7.08 (-4.44, -9.88) | -9.64 (-8.69, -10.59) | -11.04 (-8.42, -13.78) | -6.60 (-6.24, -7.07) |
| | Girls | 84.05 (21,166/ 25,182) | 74.66 (383/513) | 75.88 (3,737/ 4,925) | 74.20 (417/562) | 77.84 (53,479/68, 703) |
| | 95% CI of difference | | -9.39 (-5.79, -13.36) | -8.17 (-6.91, -9.47) | -9.85 (-6.38, -13.65) | -6.21 (-5.66, -6.76) |
| | Boys | 75.95 (20,877/ 27,488) | 70.89 (375/529) | 64.91 (3,450/ 5,315) | 63.74 (385/604) | 68.61 (48,392/ 70,530) |

*(Continued)*

**Table 2.** (Continued)

| | | Never had depression | Previous history of depression | Depression after birth | Previous history of depression & diagnosis after birth | No stable male identified |
|---|---|---|---|---|---|---|
| | 95% CI of difference | | -5.06 (-1.32, -9.10) | -11.04 (-9.67, -12.43) | -12.21 (-8.44,-16.15) | -7.34 (-6.72, -7.95) |

95% CI of difference = the reference group for confidence intervals is stable male never had depression.

Table 3 shows a similar pattern when purely adjusting for deprivation (Townsend quintile) and Table 4 illustrates potential confounding factors associated with offspring depression and educational attainment. When these were combined into a Cox regression model (Fig 1); maternal depression before the birth (HR: 1·22, 95% CI:1·11–1·33) maternal depression after the birth (HR: 1·51, 95% CI: 1·47–1·55), maternal depression before and after the birth (HR 1·73, 95% CI: 1·64–1·83), stable male depression after the birth (HR:1·43, 95% CI: (1·35–1·60), stable male depression before and after the birth (HR: 1·27, 95% CI: 1·06–1·52), were significant predictors of child depression (after adjusting for gender, deprivation, achieving key stage 2, gestation age, birth weight, number of house moves, age became mother and maternal antidepressant use). Whilst stable male depression before the birth did increase the likelihood of offspring depression in the adjusted model, this was only just statistically significant (HR:1.24, 95%CI: 1.00–1.52 (p = 0.046)).

## Educational attainment

Tables 1 and 2 show that 15·9% (35,184/220,778) of children did not achieve their KS2. 19·2% (21,762/113,275) of boys and 12·5% (13,422/107,503) of girls did not achieve their KS2. Maternal depression was associated with a lower proportion of children achieving Key stage 1, 2 and 3 (Table 1). This associated was greatest when mothers were depressed both before and after the child's birth and lowest when the mother was depressed only prior to the child's birth. For paternal depression (table 2), a lower proportion of children achieved Key stage 1, 2 and 3 when the father was depressed both prior to and after the child's birth. However, the confidence intervals overlapped for those with depression before the child was born only and the other depressed groups.

Unadjusted associations with educational attainment show in terms of parental depression, the highest risk for not attaining was a stable male or mother with depression before and after

**Table 3. Risk of depression in child if mother had depression, adjusting for deprivation (clustering on mother ID).**

| | Child depression (Adjusted Hazard ratios and 95% Confidence Intervals) | Achieving Key Stage 1 (Odds ratios and 95% Confidence intervals) | Achieving Key Stage 2 (Odds ratios and 95% Confidence intervals) | Achieving Key Stage 3 (Odds ratios and 95% Confidence intervals) |
|---|---|---|---|---|
| Mother depressed before the birth | 1.30 (1.20–1.41) | 0.74 (0.71–0.78) | 0.82 (0.77–0.87) | 0.81 (0.76–0.87) |
| Mother depressed after the birth | 1.92 (1.88–1.96) | 0.68 (0.66–0.70) | 0.71 (0.69–0.73) | 0.64 (0.62–0.66) |
| Mother depressed before and after the birth | 2.11 (2.02–2.22) | 0.59 (0.58–0.62) | 0.66 (0.63–0.69) | 0.63 (0.60–0.66) |
| Deprivation (Quintile 5-most deprived) | 1.62 (1.56–1.68) | 0.37 (0.36–0.39) | 0.36 (0.34–0.38) | 0.28 (0.27–0.29) |

*hazard ratio compared to mother with no history of depression.

**Table 4. Covariates and association with offspring depression and educational attainment.**

| | Offspring depression (Crude Hazard ratios and 95% Confidence Intervals) | Achieving KS1 (Crude Odds ratios and 95% Confidence Intervals) | Achieving KS2 (Crude Odds ratios and 95% Confidence Intervals) | Achieving KS3 (Crude Odds ratios and 95% Confidence Intervals) |
|---|---|---|---|---|
| Mother depressed before the birth | 1.32 (1.21–1.43) | 0·71 (0·68–0·74) | 0·79(0·74–0·83) | 0·77 (0·72–0·82 |
| Mother depressed after the birth | 2.00 (1.96–2.05) | 0·63 (0·61–0·65) | 0·66 (0·64–0·67) | 0·59 (0·57–0·60) |
| Mother depressed before and after the birth | 2.25 (2.15–2.35) | 0·54 (0·52–0·56) | 0·60 (0·58–0·63) | 0·57 (0·55–0·60) |
| Stable male depression before birth | 1.44 (1.18–1.74) | 0·67 (0·61–0·74) | 0·71 (0·62–0·80) | 0·67 (0·59–0·77) |
| Stable male depression after birth | 1.66 (1.58–1.74) | 0·66 (0·63–0·70) | 0·65 (0·61–0·69) | 0·60 (0·57–0·62) |
| Stable male depression before and after birth | 1.47 (1.25–1.73) | 0·55 (0·49–0·61) | 0·56 (0·50–0·64) | 0·56 (0·49–0·63) |
| No stable male identified | 1.27 (1.24–1.30) | 0·68 (0·66–0·70) | 0·74 (0·72–0·76) | 0·69 (0·67–0·71) |
| Birth weight | 0.84 (0.83–0.86) | 1·50 (1·47–1·53) | 1·46 (1·43–1·49) | 1·40 (1·37–1·42) |
| Age became mother | 0.96 (0.96–0.96) | 1·04 (1·04–1·05) | 1·04 (1·04–1·04) | 1·06 (1·06–1·06) |
| Gestation Age | 1.00 (0.99–1.00) | 1·09 (1·09–1·10) | 1·07 (1·07–1·08) | 1·05 (1·04–1·05) |
| Female | 2·05 (2·00–2·09) | 1·92 (1·88–1·97) | 1·67 (1·63–1·71) | 1·62 (1·59–1·65) |
| Number House Moves in first 5 years | 1·07 (1·06–1·08) | 0·82 (0·82–0·83) | 0·83 (0·82–0·84) | 0·77 (0·76–0·78) |
| Number infections in first 5 years | 1·08 (1·04–1·12) | 1·05 (0·98–1·15) | 0·93 (0·88–0·99) | 0·95 (0·91–0·99) |
| Achieving at age 7 (KS1) | 1·03 (0·82–1·30) | . . . | . . . | . . . |
| Achieving at age 11 (KS2) | 0·81 (0·76–0·87) | . . . | . . . | . . . |
| Achieving at age (KS3) | 0·70 (0·68–0·73) | . . . | . . . | . . . |
| Maternal anti-depressant use | 1·97 (1·93–2·02) | 0·64 (0·62–0·65) | 0·65(0·64–0·67) | 0·60 (0·59–0·61) |
| Quintile 5 (most deprived) | 1·82 (1·76–1·89) | 0.35 (0.33–0.36) | 0.34 (0.33–0.35) | 0.26 (0.25–0.27) |

the birth. When risk factors were combined using logistic regression models (Fig 2) children were less likely to achieve their key stage milestones, if either their mother or stable male within the household before and after the birth. For KS2, hazard ratios were significant if their mother/stable male had depression before the birth (HR 0.92, 95%CI:0.86, 0.99 and HR 0.85 95%CI: 0.74, 0.98) respectively), after the birth (HR 0.88, 95%CI:0.85,0.91 and HR 0.79 (95% CI:0.75,0.85)) respectively and before and after the birth (HR 0.79, 95%CI:0.75,0.84 and HR 0.74, 95%CI:0.64,0.84) respectively after adjusting for confounders. This relationship was also seen for key stage 1 (age 7) and key stage 3 (age 14).

## Discussion

These results suggest that living with a parent with depression is detrimental to a child's outcome but having a parent who has had a history of depression even prior to the birth also gives a higher risk of depression and lower educational attainment in the child. This is most pronounced with the father/stable male, where association with poor educational outcomes are similar for those who had a father with depression before the birth only and for those with

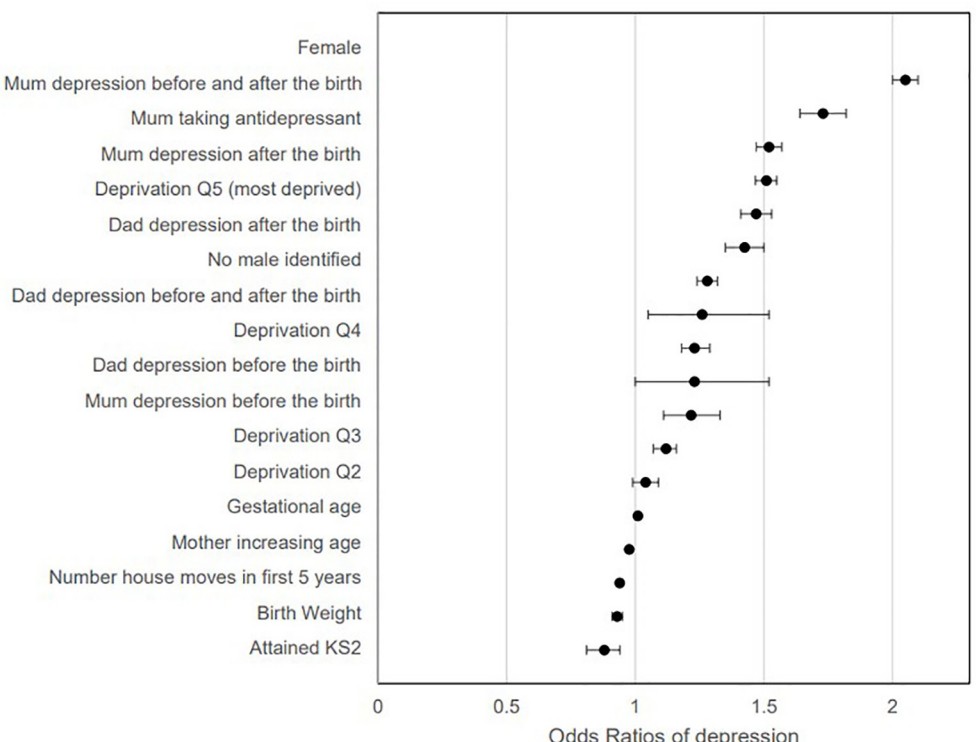

**Fig 1. Cox regression of all factors associated with depression in children.** Note: Quintile Q5 most deprived and Q1 least deprived. Attained KS2 is achieved educational milestones at age 10–11.

chronic depression. No stable male (e.g. the family have separated) was also a risk factor for depression and poor educational outcomes. The findings showed moving home (perhaps away from difficulty) was associated with less risk of depression in the child, but house moves were associated less likely to achieve educational milestones possibly through educational disruption.

The risks of developing depression were highest in offspring who were exposed to maternal depression both before and after the birth of the child. The risks of failing school exams were highest in offspring who were exposed to maternal/stable male depression before and after the birth of the child, suggesting the chronicity of depression is highly important in determining offspring outcomes. This research is in line with research to date whereby offspring whose parents are persistently depressed are at the highest risks of experiencing negative outcomes [4].

With regards to depression, the strength of association between parent and child diagnosed depression in our study was similar if the mother or stable male had depression, but with higher rates of offspring depression if the mother had chronic depression compared to if the father had chronic depression. The stronger association of depression if the child lives with a mother who has depression compared to a mother with a history of depression suggests exposure mediates child depression. This finding is in agreement with other studies in this area [28]. However, consistent with Lewis et al. [29], we also found that depression in fathers during childhood were also associated with symptoms of depression in their offspring in adolescence.

In terms of education attainment, we found that maternal and stable male depression before the child was born had a significantly negative impact on achievement. The finding that parental depression is associated with educational attainment of offspring both in the earlier

### Factors associated with achieving Key Stage 1

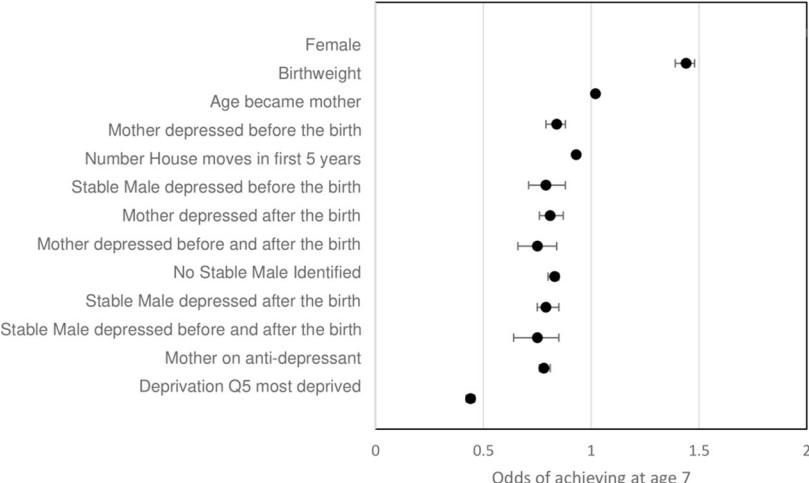

### Factors associated with achieving Key Stage 2

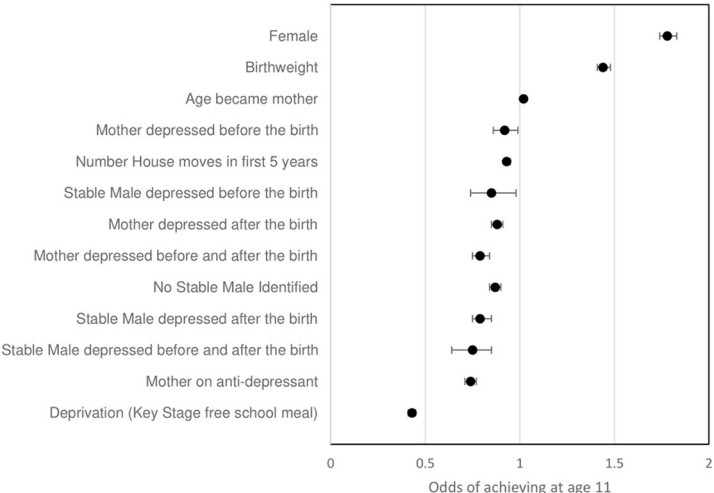

### Factors associated with achieving Key Stage 3

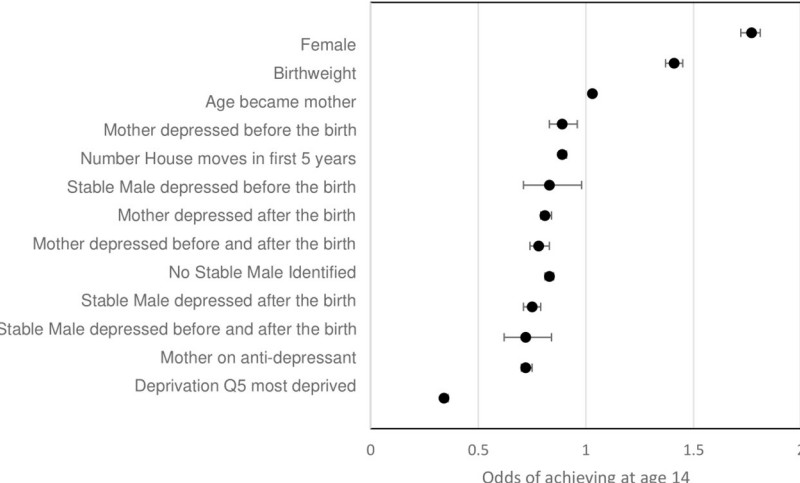

**Fig 2.** Logistic regression of all factors associated with achieving educational milestones at (a) Key stage 1 (age 7), (b) key stage 2 (age 11) and (c) key stage 3 (age 14). Note: Quintile Q5 most deprived and Q1 least deprived.

years and in the later teenage years is supported by Claessens and Shen et al respectively [3, 4]. Interestingly in our study, both maternal and stable male depression before the birth was also associated with poor academic attainment at all ages. Indeed, Shen et al [3] highlighted that genetic liability may play an important role in explaining the influence of parental depression on child outcomes as they showed maternal or paternal depression before the birth was associated with worse school performance even at age 16. However, there is still the possibility that other environmental factors such as parenting style or self confidence learned from parent may also impact on educational outcomes and this may be more pronounced when the father is the parent with a history of depression.

Shen et al [3] found paternal and maternal depression to have similar impacts on school performance in a national sample of Swedish population when based largely on inpatient diagnosis, but using a subsample of non-clinical diagnosis found the impact of maternal depression to be stronger. They suggest that the impact of less severe depression may be more detrimental when it occurs in the mother than the father highlighting the differential effect of maternal and paternal depression dependent on depression severity. The finding of stable male having a larger effect in our study in terms of educational outcomes is particularly surprising as with the linked routine data method employed in our study, we cannot be sure we have identified the biological father. We therefore expected a weaker association for the stable male compared to mothers in our analysis. However, it does highlight that support for the male living in the same household as the child if they have depression is as important in affecting child outcomes as maternal depression. It is clear that a whole family approach to mental health should be considered in order to reduce the risk of adolescent depression and poor educational attainment. Indeed, some interventions have been put in place with a specific focus on children of depressed parents with some promise but varying effects [30]. However, many of the interventions under study were based in America and wider research is needed to understand whether intervention components are easy to implement in other countries and whether the same effect is seen. Furthermore, a specific focus is required on establishing the positive components of these programmes so that this can enhance the provision provided by health visiting services, family support services and wider professionals in countries around the world.

Overall, developing depression had stronger associations with maternal depression, but educational attainment had stronger association with depression in the male of the household. Studies suggest that the mechanisms behind the associations between parental depression and offspring mental health differ depending on the mental health outcome of interest, attributing environmental mechanisms to be largely at play in offspring depression [31, 32]. The same appears to be the case with regard to educational outcomes. However, for educational outcomes, a male with diagnosed depression before the child is born is as important as the child being exposed to paternal depression. This suggests either there is a strong genetic component or that other environmental confounders such as paternal attitudes and parenting style impact on education and remain even when a father has recovered from depression.

This study highlights some important implications for practice and that a focus on environmental mechanisms could modify risk of depression and improve educational attainment for children who live with a parent with depression. Research into the environmental causes of depressive disorders identified that the absence of parental warmth, decreased parental monitoring, over-involvement, increased hostility, high inter-parental conflict, family stress, poor family functioning, maltreatment, neglect, emotional and physical abuse are all factors that may be at play in this environmental association [1, 2, 5, 33, 34]. All of these factors require whole family support. Health visiting in the United Kingdom is largely based on mothers and children, with little focus given to fathers in comparison. Indeed, a recent study highlighted a lack of coverage on paternal mental health in health visiting training, with trainees not feeling

confident in supporting fathers in their role [35]. Increased emphasis on paternal (biological or not) involvement in child development is needed to improve these outcomes. Furthermore, with our findings showing parental depression any time before or after the birth can influence offspring outcomes and perhaps maybe even have a larger effect on attainment, parental support shouldn't be limited to the early years, but ideally beyond this, with professionals (and perhaps wider) who have most contact with fathers trained to provide advice (e.g. GP's), signposting and support. Qualitative work with fathers could identify best methods of offering and accessing support and may require departure from traditional methods used in order to increase engagement of fathers. Investment in this area could reap large rewards in terms of offspring educational and health outcomes. Furthermore, given the crucial role of fathers on children's development, methods of enhancing record linkage to paternal records will give greater insight and allow us to better understand the impact of paternal factors on child outcomes.

This study possesses numerous strengths; firstly, the very large sample size employed gives some indication of wider population prevalence, with a long follow up period and formal clinical diagnosis of depression used. Findings have given insight into mechanisms which may be at play and highlighted implications for practice. However, it also has several limitations. Whilst clinical diagnosis of depression is useful, we will miss mothers and fathers who do not seek treatment for their depression. Evidence suggests that this is likely to be more pronounced for fathers as males are less likely to seek treatment for common mental illness meaning that the fathers included may represent a more chronic group. Findings for fathers can only reflect association with a stable male in the house rather than the biological father. We can only include the records of families who have registered with a GP and who register if moving. The more vulnerable families who do not register with their GP or attend their GP, will be missing from the study. There may also be unmeasured confounders outside of our data remit which have not been included for example access to childcare, parenting style, learning disability and ADHD. The hazards ratios maybe lower than reported if other possible confounders were taken into consideration during the analysis. Finally, future research would be needed to examine the exact timing of depression for each parent and the relationship with each educational attainment measure for the child.

## Conclusions

Our research highlights that parental depression is associated with depression and educational failure in children. This applies for maternal as well as paternal depression. For maternal depression, analysis to assess the impact of growing up with a depressed parent, suggests the importance of the rearing environment. For father depression, the impact on child educational failure was particularly pronounced and this shows a need to support families where depression has been present in either parent, with the impact of paternal depression requiring more attention than has been previously given. Depression is an issue that impacts on a family rather than an individual. Successfully addressing depression in a parent will also address wellbeing and potential depression in the child. Taking a holistic approach to addressing family wellbeing and depression will help ensure positive outcomes are seen in the whole family in the long term.

## Supporting information

**S1 Checklist. The RECORD statement–checklist of items, extended from the STROBE statement, that should be reported in observational studies using routinely collected health data.** (DOCX)

**S1 Table. Codes used to define depression in general practice data.**
(DOCX)

**S1 Fig. Flow diagram of inclusion and exclusion within the study.**
(TIF)

## Acknowledgments

This research was funded by the Welsh Government through the National Centre for Population Health and Wellbeing Research and makes use of anonymised data in the SAIL databank. We would like to acknowledge all the data providers who make anonymised data available for research.

## Author Contributions

**Conceptualization:** Sinead Brophy, Frances Rice.

**Data curation:** Charlotte Todd, Muhammad A. Rahman, Natasha Kennedy.

**Formal analysis:** Sinead Brophy, Charlotte Todd, Natasha Kennedy.

**Funding acquisition:** Sinead Brophy.

**Investigation:** Muhammad A. Rahman.

**Methodology:** Charlotte Todd, Frances Rice.

**Supervision:** Frances Rice.

**Writing – original draft:** Sinead Brophy, Charlotte Todd.

**Writing – review & editing:** Sinead Brophy, Charlotte Todd, Muhammad A. Rahman, Natasha Kennedy, Frances Rice.

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
