## [Decision Letter · Decision Letter 0]

22 May 2021

PONE-D-21-07577

Timing of parental depression on risk of child depression and poor educational outcomes: a population based routine data cohort study from Born in Wales, UK.

PLOS ONE

Dear Dr. Brophy,

Thank you for submitting your manuscript to PLOS ONE. After careful consideration, we feel that it has merit but does not fully meet PLOS ONE’s publication criteria as it currently stands. Therefore, we invite you to submit a revised version of the manuscript that addresses the points raised during the review process.

We look forward to receiving your revised manuscript.

Kind regards,

Yongfu Yu, Ph.D

Academic Editor

PLOS ONE

Journal Requirements:

2a) If there are ethical or legal restrictions on sharing a de-identified data set, please explain them in detail (e.g., data contain potentially identifying or sensitive patient information) and who has imposed them (e.g., an ethics committee). Please also provide contact information for a data access committee, ethics committee, or other institutional body to which data requests may be sent.

2b) If there are no restrictions, please upload the minimal anonymized data set necessary to replicate your study findings as either Supporting Information files or to a stable, public repository and provide us with the relevant URLs, DOIs, or accession numbers. Please see http://www.bmj.com/content/340/bmj.c181.long for guidelines on how to de-identify and prepare clinical data for publication. For a list of acceptable repositories, please see http://journals.plos.org/plosone/s/data-availability#loc-recommended-repositories.

Additional Editor Comments (if provided):

Reviewers' comments:

Reviewer's Responses to Questions

**Comments to the Author**

1. Is the manuscript technically sound, and do the data support the conclusions?

Reviewer #1: Yes

Reviewer #2: Yes

2. Has the statistical analysis been performed appropriately and rigorously? 

Reviewer #1: Yes

Reviewer #2: Yes

3. Have the authors made all data underlying the findings in their manuscript fully available?

Reviewer #1: Yes

Reviewer #2: Yes

4. Is the manuscript presented in an intelligible fashion and written in standard English?

Reviewer #1: Yes

Reviewer #2: Yes

5. Review Comments to the Author

Reviewer #1: Title: Timing of parental depression on risk of child depression and poor educational outcomes: a population based routine data cohort study from Born in Wales, UK

• What are the main claims of the paper and how significant are they for the discipline?

This is a well-written manuscript focusing on an important subject in child mental health and overall well-being. The research aims to show the association between maternal and paternal depression, the timing of depression in child’s life and the incidence of childhood depression and effects on their education.

• Are the claims properly placed in the context of the previous literature? Have the authors treated the literature fairly?

Discussion was good and comprehensive.

• Do the data and analyses fully support the claims? If not, what other evidence is required?

There are several other confounders that might take part in the educational outcomes of children such as any history of a learning or an intellectual disability or ADHD. The authors might not have this information as part of their dataset, but I think actual HRs might be lower than reported and perhaps even some might be statistically insignificant if other possible confounders were taken into consideration during analysis.

For the association of learning outcomes of children and maternal and paternal depression after the birth of the child, looking at the association between the timeline of the depression diagnosis and the child’s educational achievement afterwards might have been more meaningful.

On page 8, while reporting the risk effects of paternal depression groups on childhood depression, authors state that the ‘confidence intervals overlapped’. Result can still be statistically significant despite overlapping confidence intervals. Assessing confidence intervals of differences might provide additional meaningful information and decrease the type II error rate.

• PLOS ONE encourages authors to publish detailed protocols and algorithms as supporting information online. Do any particular methods used in the manuscript warrant such treatment? If a protocol is already provided, for example for a randomized controlled trial, are there any important deviations from it? If so, have the authors explained adequately why the deviations occurred?

N/A

• Are details of the methodology sufficient to allow the experiments to be reproduced?

Yes.

• Is any software created by the authors freely available?

The dataset is available online.

• Is the manuscript well organized and written clearly enough to be accessible to non-specialists?

The manuscript is well-written and easy to understand. The English and Scientific language is of adequate quality throughout the manuscript.

• Is it your opinion that this manuscript contains an NIH-defined experiment of Dual Use concern?

N/A

Reviewer #2: This is a prospective cohort data that examined the association between maternal and paternal depression, before or after the child’s birth, to depression and educational outcomes in the child. The study utilized the SAIL (Secure Anonymized Information Linkage) databank of the Welsh population that links participants in three main datasets: Welsh Demographic Service dataset, Education Attainment dataset, and Patient Episode Database. Using Cox-regression analysis, the authors observed the RR of 1.22 of offspring depression if mother had depression before child’s birth, RR of 1.55 if mother had depression after the child was born, and RR of 1.73 if she had depression before and after the child’s birth. Similarly, the RR of depression in the child if male living in the household had depression before, after, before and after are 1.24, 1.43, and 1.27 respectively.

Comments:

1. Methods. Mention the period used to define depression before and after child’s birth. If there were none, then state in the method section to give the readers some insight and application.

2. Otherwise, I found the manuscript of interest. The methodology is correct, and I have no suggestion to improve the manuscript.

6. PLOS authors have the option to publish the peer review history of their article (what does this mean?). If published, this will include your full peer review and any attached files.

Reviewer #1: No

Reviewer #2: No

---

## [Author Response · Author response to Decision Letter 0]

27 Aug 2021

Dear Editor,

Thank you very much for your letter outlining points raised during the review process. We have made the following changes: 

1. Please ensure that your manuscript meets PLOS ONE’s style requirements, including those for file naming. 

Done. 

2. a) If there are ethical or legal restrictions on sharing a de-identified data set, please explain them in detail (e.g. data contain potentially identifying or sensitive information) and who has imposed them (e.g. an ethics committee). Please also provide contact information for a data access committee, ethics committee, or other institutional body to which data requests may be sent. 

The data used in this work are provided by the SAIL (Secure Anonymised Information Linkage) Database which is a trusted third party/trusted research environment (TRE) that links identifiable data, is ISO 27001 accredited and provides data following institutional governance approval. To access SAIL data a researcher needs to apply to https://saildatabank.com/. The data used in this work cannot be removed from the SAIL gateway as it is potentially identifiable data, but access to the data can be gained by putting in an application to SAIL (https://saildatabank.com/)

3. Please include captions for your supporting Information Files at the end of your manuscript, and update any in-text citations to match accordingly. 

Done

4. Please review your reference list to ensure that is complete and correct. 

Done

Reviewers comments: 

Thank you very much for the very positive review of the paper. We have focused on the points that require an amendment or change and detailed the changes made: 

Do the data and analysis full support the claims? If not, what other evidence is required?

There are several other confounders that might take part in the educational outcome of children such as any history of learning or intellectual disability or ADHD. The authors might not have this information as part of their dataset, but I think actual HRs might be lower than reported and perhaps even some might be insignificant if other possible confounders are taken into consideration during analysis. 

We have added the following to the discussion “There may also be unmeasured confounders outside of our data remit which have not been included for example access to childcare, parenting style, learning disability and ADHD. The hazards ratios maybe lower than reported if other possible confounders were taken into consideration during the analysis”.

For the association of leaning outcomes of children and maternal and paternal depression after the birth of the child, looking at the association between the timeline of the depression diagnosis and the children educational achievement afterwards might have been more meaningful. 

We have looked at depression before the child’s birth vs depression during the lifetime of the child. The point raised here focuses on a subgroup of families where we are looking at attainment at age 7, age 11 and age 14 when the mother or the father had depression in the child’s lifetime. We feel calculating a time between depression in the mother and separately for the father/resident male and three repeated measures of for attainment would have been complicated and possibility a separate paper as it was not the main purpose of this research study.

We have added the following to the discussion: Finally, future research would be needed to examine the exact timing of depression for each parent and the relationship with each educational attainment measure for the child.

On page 8, while reporting the risk effects of paternal depression groups on childhood depression authors state that the ‘confidence intervals overlapped’. Results can still be statistically significant despite overlapping confidence intervals. 

We have added the following to the results: A comparison of those who had depression after (but not before the child’s birth) and those who had depression only before showed a HR 1.16 (95% CI: 0.95 – 1.4), showing no significant difference.

Reviewer 2: 

1. Methods: mention the period used to define depression before and after child’s birth. If there were none, then state in the methods sections to give readers some insight. 

The following has been added:

 all available GP records were used to identify depression before the birth of the child but all participants need to have a minimum of 2 years of GP records before the birth of the child. 

Thank you very much for the advice and help for improving the manuscript. 

With very good wishes and thanks

Professor Sinead Brophy on behalf of all the authors.

---

## [Editor Report · Decision Letter 1]

11 Oct 2021

Timing of parental depression on risk of child depression and poor educational outcomes: a population based routine data cohort study from Born in Wales, UK.

PONE-D-21-07577R1

Dear Dr. Brophy,

We’re pleased to inform you that your manuscript has been judged scientifically suitable for publication and will be formally accepted for publication once it meets all outstanding technical requirements.

Kind regards,

Yongfu Yu, Ph.D

Academic Editor

PLOS ONE
---

## [Editor Report · Acceptance letter]

21 Oct 2021

PONE-D-21-07577R1 

Timing of parental depression on risk of child depression and poor educational outcomes: a population based routine data cohort study from Born in Wales, UK. 

Dear Dr. Brophy:

I'm pleased to inform you that your manuscript has been deemed suitable for publication in PLOS ONE. Congratulations! Your manuscript is now with our production department. 

Kind regards, 

on behalf of

Dr. Yongfu Yu 

Academic Editor

PLOS ONE